# Metasurface orbital angular momentum holography

Haoran Ren [1,2], Gauthier Briere[3], Xinyuan Fang [1], Peinan Ni[3], Rajath Sawant[3], Sébastien Héron[3], Sébastien Chenot[3], Stéphane Vézian[3], Benjamin Damilano[3], Virginie Brändli[3], Stefan A. Maier[2] & Patrice Genevet [3]

Allowing subwavelength-scale-digitization of optical wavefronts to achieve complete control of light at interfaces, metasurfaces are particularly suited for the realization of planar phase-holograms that promise new applications in high-capacity information technologies. Similarly, the use of orbital angular momentum of light as a new degree of freedom for information processing can further improve the bandwidth of optical communications. However, due to the lack of orbital angular momentum selectivity in the design of conventional holograms, their utilization as an information carrier for holography has never been implemented. Here we demonstrate metasurface orbital angular momentum holography by utilizing strong orbital angular momentum selectivity offered by meta-holograms consisting of GaN nano-pillars with discrete spatial frequency distributions. The reported orbital angular momentum-multiplexing allows lensless reconstruction of a range of distinctive orbital angular momentum-dependent holographic images. The results pave the way to the realization of ultrahigh-capacity holographic devices harnessing the previously inaccessible orbital angular momentum multiplexing.

[1] Laboratory of Artificial-Intelligence Nanophotonics, School of Science, RMIT University, Melbourne, VIC 3001, Australia. [2] Chair in Hybrid Nanosystems, Nanoinstitute Munich, Faculty of Physics, Ludwig-Maximilians-University Munich, 80539 Munich, Germany. [3] Université Côte d'Azur, CNRS, CRHEA, rue B. Gregory, 06560 Valbonne, France. Correspondence and requests for materials should be addressed to H.R. (email: haoran.ren@physik.uni-muenchen.de) or to P.G. (email: patrice.genevet@crhea.cnrs.fr)

Metasurfaces, which allow the complete control of the wavefront of an electromagnetic wave with an ultrathin photonic device, have provided an indispensable platform for both fundamental studies of light-matter interactions[1–5] and a diverse range of photonic applications in optical microscopy and imaging[6–9], dispersion compensation[10–12], skin cloak[13], surface waves engineering[14] and multiplexing[15], and intelligent photonics[16]. Owing to the subwavelength nature of plasmonic[1] and dielectric[17–20] meta-atoms, high-resolution metasurfaces have revolutionized the photonic design of meta-holograms for holographic displays[21–24], optical encryption[25,26], and nonlinear holography[27]. In this context, meta-holograms responsive to different physical properties of light including polarization[28], helicity[29], wavelength[30], and incidence angle[31] have recently been exploited to address independent information channels for high-capacity holographic multiplexing.

Orbital angular momentum (OAM), manifested by a helical wavefront of light, has emerged as a new degree of freedom of light for boosting both optical[32,33] and quantum[34,35] information capacities. To date, however, OAM of light has not been implemented as an independent information carrier for optical holography, mainly due to the lack of OAM selectivity in conventional hologram design. Typically, a digital hologram with a quasi-continuous spatial frequency distribution destroys the extrinsic OAM of light[36], completely losing the OAM physical property in the holographic reconstruction process. Despite the fact that generation and detection of multiple wavefronts carrying the OAM have been demonstrated through holographic optical elements with only a few diffraction orders[37,38], implementing the OAM as an independent information carrier for optical holography remains elusive. More importantly, merging OAM holography with high-resolution metasurfaces could open up an unprecedented opportunity for ultrahigh-capacity holographic devices and systems, due to a physically unbounded set of OAM modes.

Here we demonstrate an entirely new concept of metasurface OAM holography capable of reconstructing a range of distinctive OAM-dependent holographic images from a single meta-hologram. We adopted subwavelength Gallium Nitride (GaN) nanopillars on a transparent sapphire substrate to digitize designed meta-holograms at a visible wavelength of 632 nm. To this purpose, three types of meta-holograms with discrete spatial frequency distributions are designed, including OAM-conserving (Fig. 1a), -selective (Fig. 1b), and -multiplexing (Fig. 1c) meta-holograms, respectively. Such a discrete spatial frequency distribution of a meta-hologram plays a key role to demonstrate the metasurface OAM holography, which preserves the OAM property in the holographic reconstruction process. In this context, an OAM-conserving meta-hologram with a discrete spatial frequency distribution is able to produce OAM-pixelated holographic images by preserving the OAM property of incident OAM beams in each pixel of reconstructed holographic images (Fig. 1a).

## Results

### OAM property preservation in metasurface holography.
According to Fourier transform holography, the spatial frequency distribution of a hologram corresponds to the electric field distribution in the image plane. Applying an incident OAM beam for the holographic reconstruction, the reconstructed electric field distribution in the image plane can thus be expressed as a convolution between a holographic image and the Fourier transform of a helical wavefront (see Supplementary Note 1). In this case, the Fourier transform of a helical wavefront, which acts as the kernel function of the convolution, is simply copied in each pixel

of the holographic image. As such, to preserve the OAM property in each pixel of a reconstructed holographic image, it is necessary to spatially sample the holographic image by an OAM-dependent two-dimensional (2D) Dirac comb function to avoid spatial overlap of the helical wavefront kernel, i.e. creating OAM-pixelated images. In this context, the constituent spatial frequencies ($k_g$ in the momentum space) of an OAM-conserving meta-hologram add a linear spatial frequency shift to an incident OAM beam ($k_{in}$). As such, outgoing spatial frequencies leaving the meta-hologram ($k_{out}$) possess a helical wavefront inherited from the incident OAM beam, which implies that the OAM-conserving meta-hologram could create OAM-pixelated holographic images (see Supplementary Fig. 1A). In contrast, previous meta-holograms[16–26] based on the conventional digital hologram design feature a quasi-continuous spatial frequency distribution that could completely destroy the helical wavefront and the associated OAM physical property of an incident OAM beam due to wave interference (see Supplementary Fig. 1B).

Mathematically, adding a spiral phase plate that features a phase distribution of $l\varphi$ ($l$ and $\varphi$ refer to the topological charge and the azimuthal angle of a phase change, respectively) on an OAM-conserving meta-hologram leads to an OAM-selective meta-hologram, of which the constituent spatial frequencies ($k_g$) carry a helical wavefront (Fig. 1b). In this case, owing to the OAM conservation, only a given OAM mode with an inverse topological charge ($-l$) can recover the fundamental spatial mode with a relatively stronger intensity distribution in each pixel of the holographic image, and hence to distinctively reconstruct the holographic image. Consequently, the OAM selectivity discussed above can be further extended to realize an OAM-multiplexing meta-hologram by superposing multiple OAM-selective meta-holograms to reconstruct a range of OAM-dependent holographic images (Fig. 1c). The latter demonstration suggests that different OAM modes can be adopted to carry independent information channels for holographic optical multiplexing. As an example, we show that incident OAM beams with topological charges of $l = -2, -1, 1,$ and $2$ can independently reconstruct distinctive holographic images (alphabet letters of A, B, C, and D) from a single OAM-multiplexing meta-hologram, respectively.

### Design and realization of an OAM-conserving meta-hologram.
The physical mechanism of metasurface OAM holography is demonstrated and illustrated in Fig. 2. To create an OAM-conserving hologram, it is necessary to sample a regular digital hologram by a 2D Dirac comb function in the spatial frequency domain, with the sampling constant ($p$) determined by the spatial frequency distribution of a spiral phase plate originating from an incident OAM beam (Fig. 2a, b and Supplementary Fig. 2). Notably, in the paraxial limit, the spatial frequency of a spiral phase plate is represented by a doughnut-shaped intensity distribution in the image plane based on the Fourier transform (see Supplementary Fig. 3). As the intensity profile of OAM modes increases with their topological charges ($l$), the sampling period ($p$) is related to $l$, the effective numerical aperture of the meta-hologram, and the wavelength of light, respectively (see Supplementary Note 1). Without the loss of generality, OAM-dependent sampling constants ($p$) were numerically characterized by calculating a hologram with an effective numerical aperture of 0.05 at a wavelength of 632 nm.

OAM-conserving meta-hologram have been experimentally realized, with the use of subwavelength GaN nanopillars with a fixed height of 1 μm and various radii for the phase-only digitalization. To this purpose, the transmission efficiency and phase retardation of light scattered from GaN nanopillars were numerically characterized using finite difference time domain

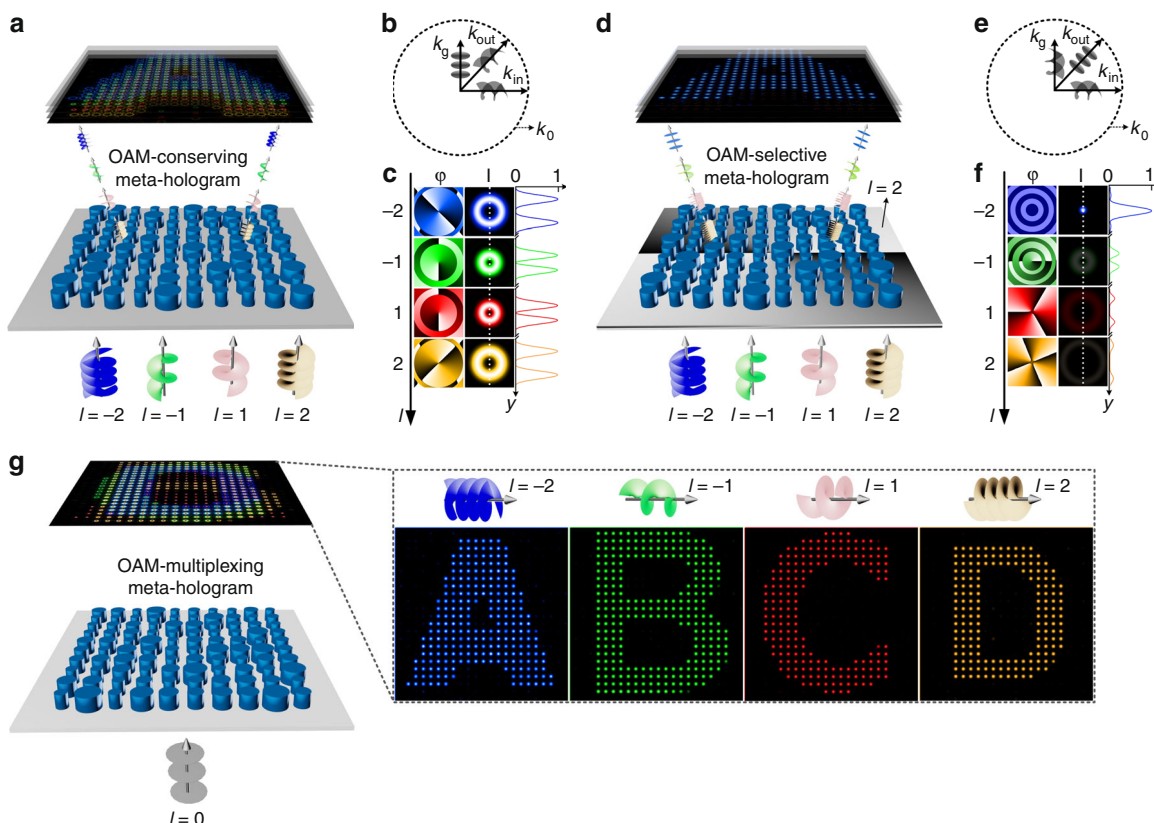

**Fig. 1** Principle of metasurface OAM holography. **a** Schematic of an OAM-conserving meta-hologram capable of transferring the OAM property from an OAM incident beam to a holographic image. **b** Schematic illustration of the OAM property transfer in the spatial frequency domain (*k*-space). **c** The phase (*φ*) and intensity (*I*) distributions of single pixels in the reconstructed holographic images, respectively. Pseudo colors are used to visualize different OAM modes. **d** Schematic of an OAM-selective meta-hologram sensitive to a given OAM mode. **e** Schematic illustration of the OAM conversion from an incident OAM beam to a fundamental spatial mode after passing through an OAM-selective meta-hologram. **f** The phase (*φ*) and intensity (*I*) distributions of single pixels in the reconstructed holographic images using different OAM modes. High intensity is achieved in each pixel whenever the incident light has a topological charge matching the design of the interface. **g** Schematic of an OAM-multiplexing meta-hologram capable of reconstructing multiple distinctive OAM-dependent holographic images

(FDTD) simulations as a function of the nanopillars radius at a wavelength of 632 nm, respectively (Fig. 2c). Obviously, the GaN nanopillars exhibit high transmission efficiency as well as the complete phase modulation ([0, 2π]), which is required for high diffraction efficiency meta-holograms. In our experiment, we selected five different sized nanopillars with radii of 76, 80, 86, 94, and 104 nm to design and fabricate a five-level OAM-conserving meta-hologram (see Supplementary Note 2). To achieve the lensless reconstruction of a holographic image directly from a meta-hologram, the phase function of a Fourier transform holographic lens is complemented in the design of an OAM-conserving meta-hologram (see Supplementary Fig. 4). Consequently, OAM-conserving meta-holograms with a physical size of 200 μm by 200 μm and a lattice constant of 340 nm by 340 nm were experimentally fabricated (Fig. 2d), which feature an effective numerical aperture of 0.05 (see 'Methods' and Supplementary Fig. 5).

Devices based on the OAM-conserving meta-holograms, providing lensless reconstruction of OAM-carrying holographic images based on different incident OAM beams, were experimentally characterized (Fig. 2e). Specifically, we show that OAM beams with topological charges of $l = \pm 1$ and $\pm 2$ can be used to reconstruct OAM-carrying holographic images from meta-holograms with sampling constants of $p = 19.8$ μm and $p = 26.1$ μm, respectively. In the experiment, a spatial light modulator (see 'Methods' and Supplementary Fig. 6) was used to generate incident OAM beams that were thereafter weakly focused and

spatially aligned with meta-holograms (see Supplementary Fig. 7). The experimental results suggest that incident OAM beams could impart the OAM property to the entire holographic images, as revealed by the doughnut-shaped intensity distributions and the interference patterns in each pixel of the reconstructed holographic images, respectively (see Supplementary Fig. 8). A small intensity fluctuation in the OAM pixels of holographic images might originate from three factors: an insufficient phase modulation due to imperfect nanofabrication, small optical aberrations in the metasurface imaging system, and the nonuniform photon sensitivity by a pixelated charge-coupled device camera.

**OAM selectivity in metasurface holography.** To achieve OAM selectivity in metasurface holography, the phase function of a spiral phase plate with a topological charge of *l* was further added onto the design of an OAM-conserving meta-hologram, leading to an OAM-selective meta-hologram (Fig. 3a). In this context, holographic images appear only when an OAM beam with an inverse topological charge of –*l* is incident on an OAM-selective meta-hologram. We designed and fabricated four OAM-selective meta-holograms (labelled as "1", "2", "3", and "4") to experimentally confirm strong OAM selectivity (Fig. 3b). As a result, different holographic images can be selectively reconstructed from the OAM-selective meta-holograms based on the incident OAM beams with topological charges of −2, −1, 1, and 2,

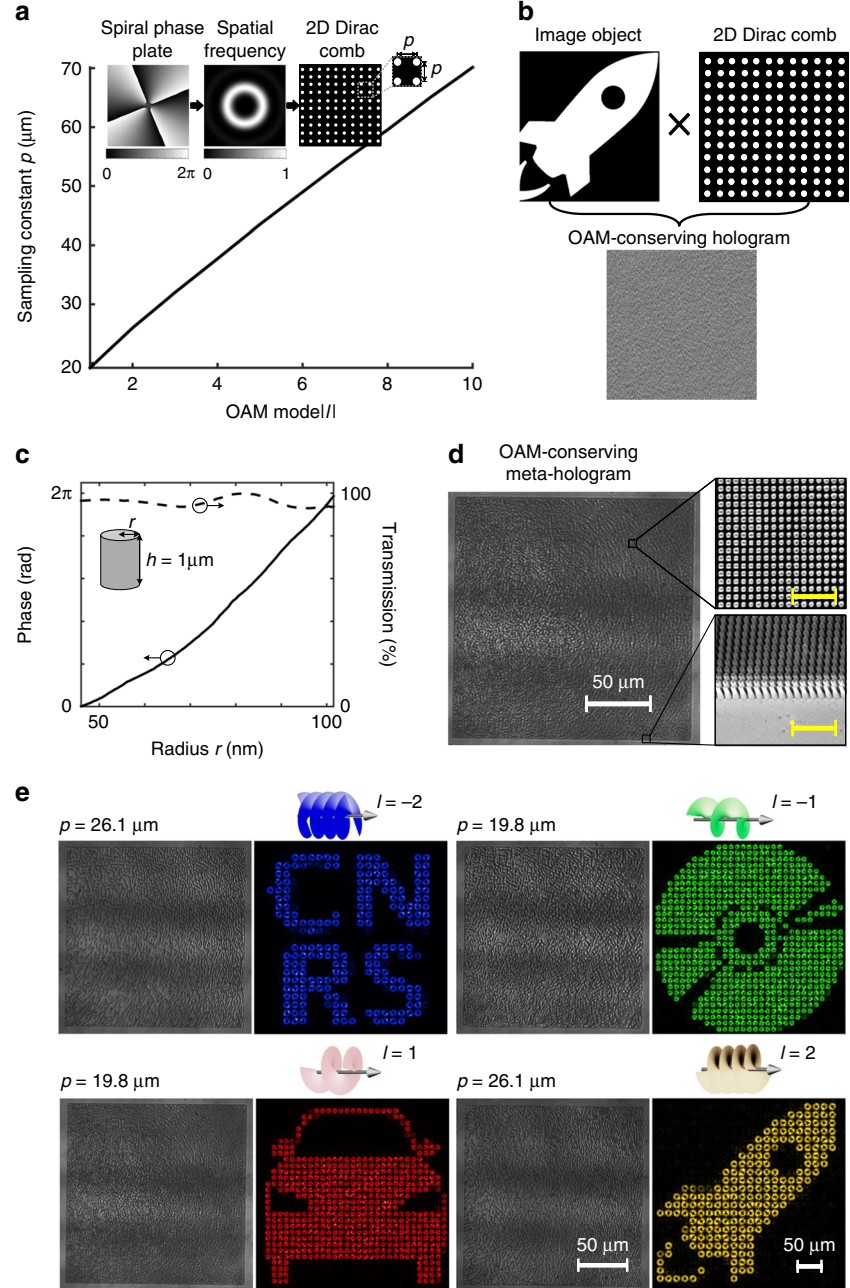

**Fig. 2** Design and fabrication of an OAM-conserving meta-hologram. **a** Numerical characterization of OAM-dependent spatial frequency sampling of an OAM-conserving meta-hologram. Insets show that a spatial frequency analysis of a spiral phase plate that is used for the generation of an incident OAM beam leads to a doughnut-shaped intensity distribution in the image plane, which further determines the sampling constant ($p$) of an OAM-conserving meta-hologram. Avoiding interference between OAM pixels in the image plane is required to maintain the OAM information. This indicates that the image sampling size has to increase with the OAM topological charges. **b** Schematic design of an OAM-conserving meta-hologram through multiplying an object image with an OAM-dependent 2D Dirac comb function (a constant periodicity: $p$) in the image plane. **c** Characterization of transmission efficiency and phase response of GaN nanopillars as a function of nanopillars radius at a wavelength of 632 nm, respectively. **d** The optical image of a fabricated OAM-conserving meta-hologram, where the top- and oblique-view scanning electron microscopy images of enlarged areas in the meta-hologram are presented. The scale bar in the inset is 2 μm. **e** Experimental characterization of the lensless reconstruction of OAM-carrying holographic images from different OAM-conserving meta-holograms through using incident OAM beams with topological charges of $l = \pm 1$ and $\pm 2$, respectively. The original "disc", "car", and "rocket" images were obtained from Flaticon, Free Icons Library, and Interactivecoding websites, respectively

respectively. Such strong OAM selectivity (Fig. 3c) originates from both the spatial and intensity distinctions between a fundamental spatial mode and high-order OAM modes in the image plane. Each pixel in the selectively reconstructed holographic images features a fundamental spatial mode dictated by a solid-spot intensity distribution. To further improve the OAM selectivity, a fundamental mode filtering aperture array in the detector plane was added to rule out high-order OAM modes with doughnut-shaped intensity distributions (see Supplementary Fig. 9). To realise a fundamental mode filtering aperture array in the experiment, the photon sensitivity of individual pixels of a charge-coupled device was spatially adjusted in accordance with

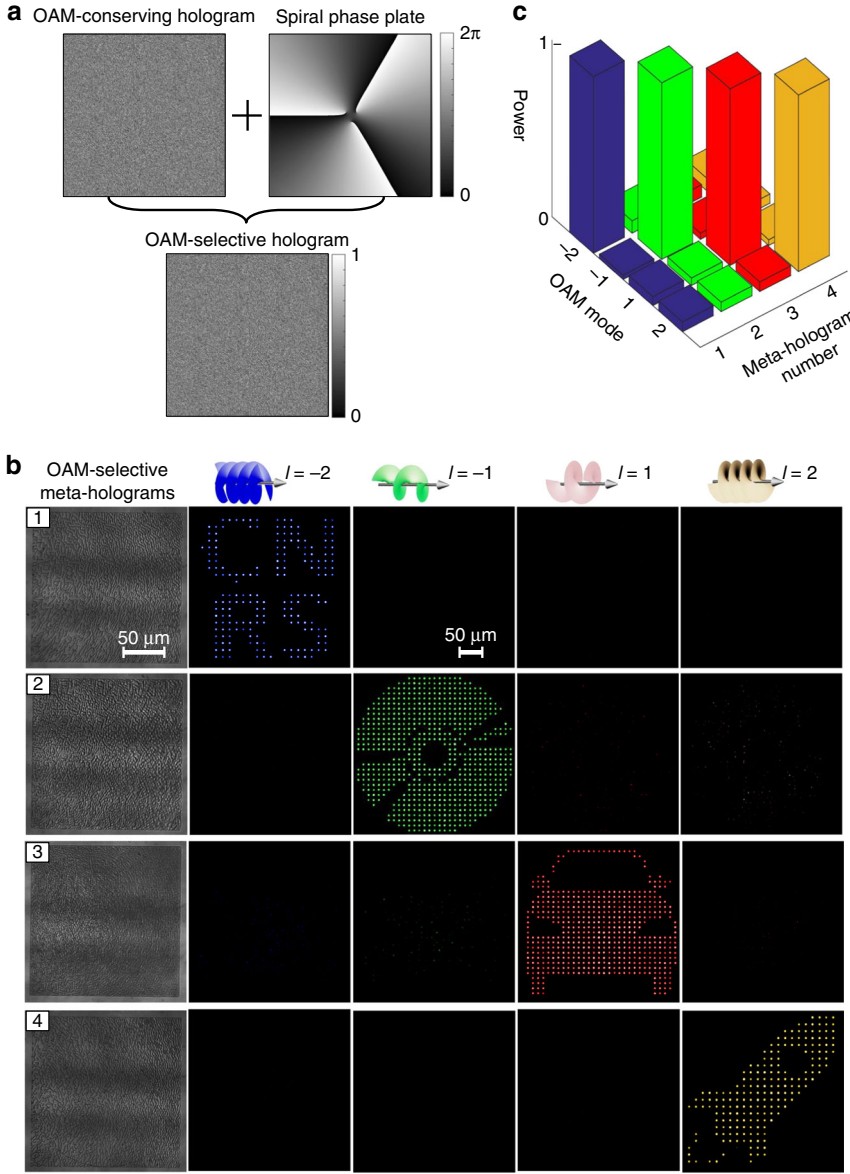

**Fig. 3** Design and characterization of OAM-selective meta-holograms. **a** Design principle of an OAM-selective hologram by adding the phase function of a spiral phase plate onto an OAM-conserving hologram. **b** Experimental characterization of OAM-selective meta-holograms, which are labelled as "1", "2", "3", and "4", based on the illumination of the four OAM beams with topological charges of −2, −1, 1, and 2, respectively. **c** Comparison of optical power of the reconstructed holographic images using different OAM beams, which suggests the strong OAM selectivity by different OAM-selective meta-holograms

OAM-dependent sampling constants. Additionally, the impact of misalignment of an incident OAM beam with respect to an OAM-selective hologram was numerically characterized, which suggests that the OAM selectivity could be maintained when their spatial misalignment is smaller than $H/4$, where $H$ represents the physical size of a meta-hologram (see Supplementary Fig. 10).

**Characterization of an OAM-multiplexing meta-hologram.** Based on the discovered OAM sensitivity, we can achieve the later holographic optical multiplexing of multiple OAM-dependent holographic images. The multiplexing approach is schematically illustrated in Fig. 4a, where four image objects (alphabet letters "A", "B", "C", and "D") were sampled in the spatial frequency domain, and their resultant OAM-selective holograms were encoded with spiral phase plates with topological charges of $l = 2$, 1, −1, and −2, respectively. Superposing the four OAM-selective holograms leads to the design of an OAM-multiplexing meta-

hologram (see Supplementary Note 3 and Fig. 4b). As a result, an incident beam with a planar wavefront could reconstruct a complex interference pattern from the OAM-multiplexing meta-hologram (Fig. 4c). On the other hand, OAM beams with topological charges of $l = −2, −1, 1$, and 2 could unambiguously reconstruct four distinctive OAM-dependent holographic images from the OAM-multiplexing meta-hologram, respectively (Fig. 4d). Notably, owing to the use of a fundamental mode filtering array, OAM-dependent holographic images, even with spatial overlap, can be reconstructed with a high fidelity through different incident OAM beams. In addition, we demonstrate the design of a 10-bit OAM-multiplexing meta-hologram for the massive reconstruction of up to $2^{10}$ OAM-dependent holographic images with a high signal-to-noise ratio (see Supplementary Fig. 11), opening the possibility of using the OAM degree of freedom for ultrahigh-capacity holographic multiplexing (see Supplementary Fig. 12) and OAM encryption.

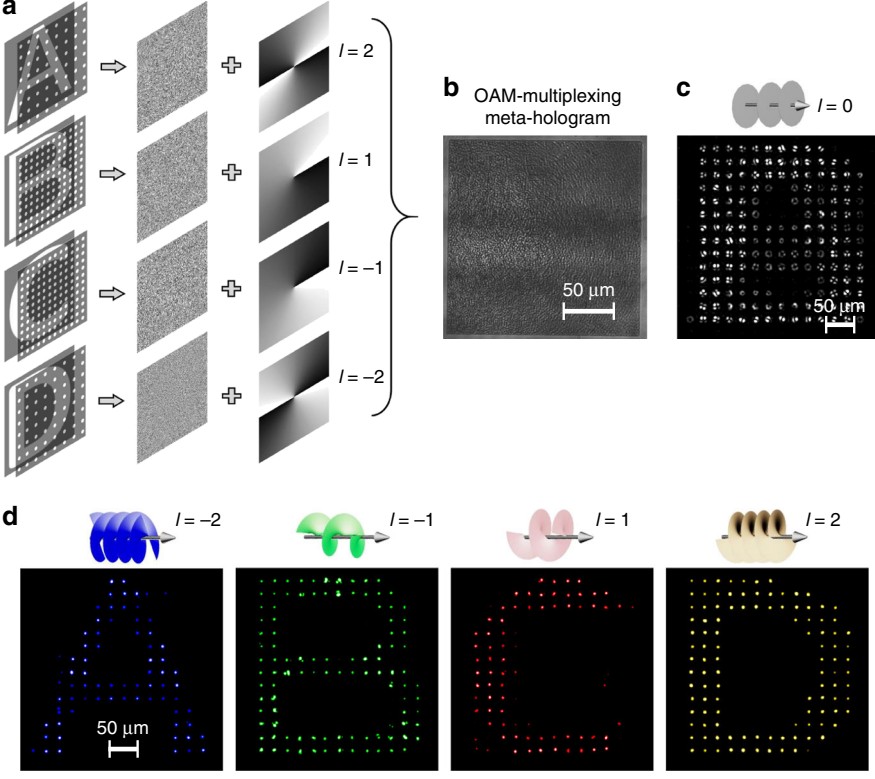

**Fig. 4** Design and experimental characterization of an OAM-multiplexing meta-hologram capable of the lensless reconstruction of multiple distinctive OAM-dependent holographic images. **a** The design approach of an OAM-multiplexing hologram. **b** The optical image of a fabricated OAM-multiplexing meta-hologram based on the design in **a**. **c** The reconstruction of a complex interference pattern from the OAM-multiplexing meta-hologram through an incident beam with a planar wavefront. **d** Experimental reconstruction of four distinctive OAM-dependent holographic images through incident OAM beams with topological charges of $l = -2, -1, 1, 2$, respectively

## Discussion

Our demonstration provides a holographic paradigm shift for harnessing the previously inaccessible OAM-division multiplexing as an independent information carrier in metasurface holography. Owing to mathematically orthogonal OAM modes without a topological charge limit, a large number of OAM-dependent information channels can therefore be multiplexed by a single meta-hologram with high spatial-resolution, which holds great promise for ultrahigh-capacity holographic devices and systems. Notably, the reported OAM-selective meta-holograms feature a high efficiency in the visible range (see Supplementary Note 4 and Supplementary Fig. 13). Moreover, a large effective mode index change as a function of radius opens the possibility of a strong phase shift offered by GaN nanopillars, laying the physical foundation of the strong phase sensitivity to the radius of GaN nanopillars (see Supplementary Note 5 and Supplementary Fig. 14). In addition, resonant modes could be slightly influenced by the fabricated GaN nanopillars with a tapered side wall[39]; however, the discrepancy in their relatively phase changes between nanopillars with a straight and tapered wall is small (see Supplementary Fig. 15). As such, this tapering effect in the fabricated GaN nanopillars exhibits a negligible influence on the meta-hologram performance. Even though OAM-dependent pixel size in the holographic image is dependent on the numerical aperture of a Fourier transform holographic lens incorporated into our meta-hologram design, for practical holographic display application this relatively large OAM pixel would not degrade the image quality of a holographic image, due to the fact that human naked eyes with regular vision cannot distinguish objects smaller than about 100 μm. On the other hand, high spatial-resolution

metasurface holograms are critical for practical holographic applications, which could lead to holographic images with a larger viewing angle, a reduced multiplexing crosstalk, and a higher diffraction efficiency. Therefore, it is highly beneficial to implement OAM holography on high-resolution metasurfaces.

Scaling high-resolution meta-holograms into a large area through direct laser printing[40] and nanoimprinting techniques[41] could truly open up new perspectives for OAM-dependent holographic applications, such as multicasting holographic display and ultrahigh-security holographic encryption. It might also be interesting from both fundamental and application points of view to extend metasurface OAM holography into the nonlinear regime. The OAM-multiplexing meta-hologram can be further implemented by active metasurfaces for dynamic optical components[25,42–47], paving the way for a wide range of applications including three-dimensional display[48,49], digital holographic microscopy and imaging[50], holographic optical trapping[51], and all-optical machine learning and artificial intelligence[16,52,53].

## Methods

**Design of GaN meta-holograms**. Meta-holograms that were used for the demonstration of metasurface orbital angular momentum (OAM) holography were realized by disposing Gallium Nitride (GaN) nanopillars with designed structural parameters to introduce proper forward scattering phase and amplitude at defined positions along the interface. The amplitude and phase responses are related to the radius of nanopillar meta-atoms with a constant height of 1 μm. The subwavelength lattice constant in meta-holograms is 340 nm, which is sufficiently small to avoid the diffraction effect in both air and substrate. To quantify the phase retardation of light transmitted through GaN nanopillars, electromagnetic simulations of subwavelength nanopillars arranged in a square lattice were performed using the FDTD. We specify the dispersion of GaN nanopillars from ellipsometry measurements realized on epitaxially grown GaN thin-film on a double-side

polished sapphire wafer. In the FDTD simulation, perfectly matched layer (PML) conditions in the direction of the light propagation and periodic boundary conditions along all the in-plane directions were used, respectively. The use of PML boundary conditions in the propagation direction results in an open space simulation while in-plane periodic boundary conditions mimic a subwavelength array of identical nanostructures. As a result, the transmission efficiency and the phase response of GaN nanopillars as a function of the radius were numerically characterized, as shown in the Fig. 2c.

**Fabrication of GaN meta-holograms**. The nanofabrication of meta-holograms was realized by patterning a 1 μm thick GaN thin-film grown on a double-side polished c-plan sapphire substrate via a Molecular Beam Epitaxy (MBE) RIBER system. The GaN nanopillars were fabricated using a conventional electron beam lithography system (Raith ElphyPlus, Zeiss Supra 40) process with metallic Nickel (Ni) hard masks through a lift-off process. To this purpose, a double layer of around 200 nm PMMA resists (495A4 then 950A2) was spin-coated on the GaN thin-film, prior to baking the resist at a temperature of 125 °C. E-beam resist exposition was performed at 20 keV. Resist development was realized with 3:1 IPA: MIBK and a 50 nm thick Ni mask was deposited using E-beam evaporation. After the lift-off process in the acetone solution for 4 h, GaN nanopillar patterns were created using reactive ion etching (RIE, Oxford system) with a plasma composed of $Cl_2CH_4Ar$ gases. Finally, the Ni mask on the top of GaN nanopillars was removed by using chemical etching with 1:2 solution of HCl: $HNO_3$.

**Optical setup**. The primary optical components used for characterizing the OAM-conserving, -selective, and -multiplexing meta-holograms are shown in Supplementary Fig. 5. A laser beam at a wavelength of 632 nm propagates through a broadband linear polarizer (10GT04, Newport). The linearly-polarized beam was dynamically modulated by a spatial light modulator (Hamamatsu X13138–01) to imprint a helical wavefront onto the optical beam via the phase function of a spiral phase plate. After that, the incident OAM beams were relayed to the meta-hologram sample surface through an achromatic 4f telescope. Thereafter, the incident OAM beams were weakly focused by an achromatic lens (L1) with a focal length of 50 mm onto a meta-hologram, which was mounted on a three-dimensional translation stage. The reconstructed holographic image from a meta-hologram sample was collected by a transmissive objective lens (L2) with a numerical aperture of 0.3 and a tube lens (L3) with a focal length of 200 mm and imaged on a pixelated charge-coupled device camera (Thorlabs, DCU223M).

## Data availability
The data that support the plots within this paper and other findings of this study are available from the corresponding authors upon reasonable request.

## Code availability
The code used for the meta-hologram design is available from the corresponding author upon reasonable request.

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

## Acknowledgements

H.R. acknowledges the funding support from the Victoria Fellowship as well as the Humboldt Research Fellowship from the Alexander von Humboldt Foundation. P.G., G.B., P.N., S.H., S.C., and V.B. acknowledge funding from the European Research Council (ERC) under the European Union's Horizon 2020 research and innovation programme (Grant agreement no. 639109).

## Author contributions

H.R. and P.G. proposed the idea and conceived the experiment; H.R. performed the calculation of meta-holograms, constructed the optical characterization of meta-holograms, and acquired the data; P.G. and G.B. contributed to the GaN nanopillars simulation and metasurface design; G.B., P.N., S.H., S.C., and V.B. contributed to the nanofabrication; X.Y. contributed to the numerical simulation; S.V. and B.D. contributed to the GaN MBE growth; H.R., S.A.M., and P.G. contributed to the data analysis; all the authors completed the writing of the manuscript.
