## [Peer Review File · Nature Communications]

Editor Note: This manuscript has been previously reviewed at another journal that is not operating a transparent peer review scheme. This document only contains reviewer comments and rebuttal letters for versions considered at Nature Communications. Mentions of the other journal have been redacted.

Reviewers' Comments:

Reviewer #1:

Remarks to the Author:

I am happy with the authors' explanation and clarification. The authors added new figure Fig. S10(C), which addressed the ratio of signal-to-noise very well. As I wrote before, the idea is interesting and the results are convincing. The paper is well written, and all display items are very nicely presented. I believe this manuscript will be of interest to specialists within the field of optics, which is covered by Nature Communications. Thus, it seems to me that the manuscript can be published in Nature Communications after one minor amendment.

In the experimental results shown in Figure 2E, Figure 3B, Figure 4D, the intensity fluctuation in the OAM pixels can be clearly observed. Analysis of the intensity fluctuation (minimum and maximum) based on the collected images and several sentences of explanation should be provided.

Reviewer #2:

Remarks to the Author:

I think this manuscript can be accepted for publication in Nature Communications since all my comments have been addressed.

Reviewer #3:

Remarks to the Author:

The authors comprehensively response all of my 8 comments, all the answers and additional results and supplements are fine. I would like to suggest to accept this paper for publication in Nature Communications.

Response to Reviewers

Reviewer #1 (Remarks to the Author):

I am happy with the authors' explanation and clarification. The authors added new figure Fig. S10(C), which addressed the ratio of signal-to-noise very well. As I wrote before, the idea is interesting and the results are convincing. The paper is well written, and all display items are very nicely presented. I believe this manuscript will be of interest to specialists within the field of optics, which is covered by Nature Communications. Thus, it seems to me that the manuscript can be published in Nature Communications after one minor amendment.

In the experimental results shown in Figure 2E, Figure 3B, Figure 4D, the intensity fluctuation in the OAM pixels can be clearly observed. Analysis of the intensity fluctuation (minimum and maximum) based on the collected images and several sentences of explanation should be provided.

Reviewer #2 (Remarks to the Author):

I think this manuscript can be accepted for publication in Nature Communications since all my comments have been addressed.

Reviewer #3 (Remarks to the Author):

The authors comprehensively response all of my 8 comments, all the answers and additional results and supplements are fine. I would like to suggest to accept this paper for publication in Nature Communications.

Reply: We thank all the reviewers for the endorsement of our revision and the recommendation for publishing our results in *Nature Communications*.

Regarding the minor comment from Reviewer 1, we have analyzed the intensity fluctuation in OAM pixels in Figures 2E, 3B, and 4D, as shown in the figure below. One sentence of explanation on the intensity fluctuation has been added in the main text (Paragraph 2, Page 6).

“A small intensity fluctuation in the OAM pixels of holographic images might originate from three factors: an insufficient phase modulation due to imperfect nanofabrication, small optical aberrations in the metasurface imaging system, and the nonuniform photon sensitivity by a pixelated charge coupled device camera.”